# Insights from a Multi-Omics Integration (MOI) Study in Oil Palm (*Elaeis guineensis* Jacq.) Response to Abiotic Stresses: Part One—Salinity

**DOI:** 10.3390/plants11131755

**Published:** 2022-06-30

**Authors:** Cleiton Barroso Bittencourt, Thalliton Luiz Carvalho da Silva, Jorge Cândido Rodrigues Neto, Letícia Rios Vieira, André Pereira Leão, José Antônio de Aquino Ribeiro, Patrícia Verardi Abdelnur, Carlos Antônio Ferreira de Sousa, Manoel Teixeira Souza

**Affiliations:** 1Graduate Program of Plant Biotechnology, Federal University of Lavras, Lavras 37200-000, Brazil; cleiton_court@hotmail.com (C.B.B.); ThallitonS@gmail.com (T.L.C.d.S.); leticia_rios1518@hotmail.com (L.R.V.); 2Embrapa Agroenergia, Brasília 70770-901, Brazil; jorgecrn@hotmail.com (J.C.R.N.); andre.leao@embrapa.br (A.P.L.); jose.ribeiro@embrapa.br (J.A.d.A.R.); patricia.abdelnur@embrapa.br (P.V.A.); 3Embrapa Meio Norte, Teresina 64006-245, Brazil; carlos.antonio@embrapa.br

**Keywords:** transcriptomics, proteomics, metabolomics, integratomics, abiotic stress, African oil palm

## Abstract

Oil palm (*Elaeis guineensis* Jacq.) is the number one source of consumed vegetable oil nowadays. It is cultivated in areas of tropical rainforest, where it meets its natural condition of high rainfall throughout the year. The palm oil industry faces criticism due to a series of practices that was considered not environmentally sustainable, and it finds itself under pressure to adopt new and innovative procedures to reverse this negative public perception. Cultivating this oilseed crop outside the rainforest zone is only possible using artificial irrigation. Close to 30% of the world’s irrigated agricultural lands also face problems due to salinity stress. Consequently, the research community must consider drought and salinity together when studying to empower breeding programs in order to develop superior genotypes adapted to those potential new areas for oil palm cultivation. Multi-Omics Integration (MOI) offers a new window of opportunity for the non-trivial challenge of unraveling the mechanisms behind multigenic traits, such as drought and salinity tolerance. The current study carried out a comprehensive, large-scale, single-omics analysis (SOA), and MOI study on the leaves of young oil palm plants submitted to very high salinity stress. Taken together, a total of 1239 proteins were positively regulated, and 1660 were negatively regulated in transcriptomics and proteomics analyses. Meanwhile, the metabolomics analysis revealed 37 metabolites that were upregulated and 92 that were downregulated. After performing SOA, 436 differentially expressed (DE) full-length transcripts, 74 DE proteins, and 19 DE metabolites underwent MOI analysis, revealing several pathways affected by this stress, with at least one DE molecule in all three omics platforms used. The Cysteine and methionine metabolism (map00270) and Glycolysis/Gluconeogenesis (map00010) pathways were the most affected ones, each one with 20 DE molecules.

## 1. Introduction

Oil palm (*Elaeis guineensis* Jacq.) has the highest productivity among the main oilseed crops, reaching 3–8 times more oil per area than any other crop [1]. In 2021/2022, approximately 82 million metric tons of palm oil and palm kernel oil was consumed worldwide, making oil palm the number one source of consumed vegetable oil in the world [2]. It is the raw source of a wide range of products used by many industries, such as the food and steel industries, the manufacture of cleaning products, the pharmaceutical and cosmetics industries, and the biofuels industry [3].

Several countries placed, in the equatorial belt, expanded oil palm plantations in tropical forests where this crop meets its natural condition of high rainfall throughout the year [4]. Despite the significant economic gains, this movement imposes environmental stresses, such as biodiversity loss, greenhouse gas emissions, land degradation, and air and water pollution [1]. In Brazil, over 95% of the oil palm plantations are in the Amazon rainforest, where only 2.14% of the total area destined for commercial plantations is currently in use [5]. This under-utilization status is due to many constraints, such as environmental legal restrictions imposed by the Brazilian Government and logistical difficulties, which together hinder the production flow to the main industrial areas in the country and also the occurrence of pests and diseases [6,7].

Outside the Amazon rainforest, there is an extensive area in Brazil with favorable conditions for cultivating oil palm [8]. There are many logistic advantages to producing oil palm outside the Amazon region, offering a window of opportunity to increase the area with oil palm plantations and, consequently, the total national palm oil yield. However, these areas experience long periods of drought throughout the year when oil palm does not meet the physiological water requirement to maintain productivity [8,9,10]. Consequently, the oil palm grower needs to irrigate the crop and must do so with proper management to avoid soil salinization.

Approximately 30% of the world’s irrigated agricultural lands are damaged by salinity, negatively affecting the productivity of traditional crops [11]. Most crop plants have evolved under very low soil salinity, and, under high salt, their development is highly inhibited, even leading to death at a concentration ranging between 100 and 200 mM NaCl [12]. Salinity stress affects plants by decreasing the osmotic potential of the soil solution, making it harder for the root to absorb water from the soil and consequently experiencing drought stress, and by accumulating sodium and chloride ions in the cytoplasm, leading to the inhibition of many enzyme reactions due to ion toxicity [13]. Salt stress tolerance in plants involves many morphophysiological and biochemical changes, such as stomatal closing, osmolyte accumulation, and increased Na^+^/Cl^−^ antiporter, governed by multigenic traits [14].

Considering those circumstances, it is clear that any initiative to promote oil palm cultivation outside the Amazon Forest in Brazil needs to take drought and salinity together when researching for knowledge and technology to empower breeding programs to develop superior genotypes for those regions. The first challenge is understanding the morphophysiological, biochemical, and molecular responses of oil palm to these two abiotic stresses. In doing so, our group has studied the response of young oil palm plants from different angles, applying different omics platforms, alone or in combination [6,15,16]. Vieira and colleagues showed that young oil palm submitted to a high concentration of NaCl reduces the rates of CO_2_ assimilation, stomatal conductance to water vapor, and transpiration [6]. Then, Ref. [15] confirmed a preponderant role of transcription factors in the early response of oil palm plants to salinity stress, and [16] identified the pathways and the metabolites most affected by drought stress.

The current study is a new step on our research activities characterizing the biochemical and molecular responses of *E. guineensis* to salinity stress, where we carried out a comprehensive, large-scale, single-omics analysis (SOA), and Multi-Omics Integration (MOI) analysis of the metabolome, transcriptome, and proteome profiles on the leaves of young oil palm plants submitted by Vieira and colleagues [6] with repsect to very high salinity stress.

## 2. Results

### 2.1. Oil Palm Transcriptome under Salinity Stress

When comparing salt-stressed against control plants, the pairwise differential expression analysis revealed 2728 differentially expressed (DE) proteins in the oil palm genome at False Discovery Rate (FDR) ≤ 0.05 in which 1138 were upregulated (Log_2_(FC) > 0) and 1590 were downregulated (Log_2_(FC) < 0) (Table 1, Appendix A).

A total of 1165 proteins with 792 distinct K numbers were present among the 2728 DE ones, including 693 enzymes, from which 436 belonged to known pathways (Appendix A).

The set of 693 enzymes underwent gene ontology analyses, and only the ten most populated groups per GO term are shown in Figure 1. The biological process subgroups with the largest number of representatives were carbohydrate metabolic process, followed by protein phosphorylation and fatty acid biosynthesis process. For molecular function, the most populated subgroups were ATP binding, metal ion binding, and heme binding. Finally, for cellular component the integral component of membrane came in first, followed by cytoplasm and cytosol components.

Furthermore, the DE enzymes were also classified according to the Enzyme Commission (EC) number, a numerical classification scheme for enzymes based on the chemical reaction. At a first level of classification that involves a general type of enzyme-catalyzed reaction that ranges from one to six, enzymes were dominated by oxidoreductases (EC 1), transferases (EC 2) and hydrolases (EC 3) classes (Figure 2a). In the subclasses of oxidoreductases class (EC 1), DE enzymes were represented mainly by those acting on paired donors, with the incorporation of or reduction in molecular oxygen (EC 1.14), followed by enzymes acting on the CH-OH group of donors (EC 1.1) and acting on the aldehyde or oxo group of donors (EC 1.1). The most representative subclass of transferases class (EC 2) included those with a function of transferring phosphorus-containing groups (EC 2.7), acyltransferases (EC 2.3) and glycosyltransferases (EC 2.4). Finally, the hydrolases (EC 3) had subclasses with compounds involved and acting on ester bonds (EC 3.1), glycosylases (EC 3.2), and acting on peptide bonds (peptidases) (EC 3.4) subclasses that were standing out.

### 2.2. Oil Palm Proteome under Salinity Stress

A global proteomics analysis led to the identification of 3234 and 2872 peptides with high confidence (FDR ≤ 0.01) in control and stressed samples, respectively, which infers up to 1809 protein entries from *E. guineensis* proteome (Uniprot) in both conditions—control and stressed (Table 2).

Approximately 38% of proteins (688) were inferred from more than four peptides, and about 34% (622) had at least one proteotypic peptide observation. A list of all peptides and proteins confidently identified, as well as a simplified list of 1019 proteins according to the maximum parsimony criterion, is presented in Appendix A. Control and stressed conditions shared 662 protein identifications; 62 and 89 proteins were uniquely detected in control and stressed samples, respectively (Figure 3a, Appendix A).

Twenty proteins showed statistically significant differences in their abundance between stressed and control samples (Table 3). As shown in Figure 3b, 12 proteins (in blue) were significantly up-regulated while eight (in red) were significantly down-regulated between stressed and control samples. A group of 642 proteins did not meet the statistical criteria and was not considered for further analysis (Table 1). Our differential abundance analysis considered proteins identified at least in two replicates in each condition. This filtering process decreased the list of 316 and 380 proteins uniquely identified in stressed and control to 89 and 62, respectively.

This group of 171 DE protein sequences—including those found exclusively in control (62) and stressed (89) in at least two replicates and those 20 proteins that attended to the statistical criteria of PatternLab V software—was submitted to functional annotation and MOI analyses. The KEGG mapper reconstruction results revealed 131 proteins with 84 distinct K numbers, including 99 enzymes. Seventy-three enzymes belonged to known pathways and were used in the MOI analysis.

This set of 171 selected proteins was then submitted to gene ontology analyses, and again only the ten most populated groups per GO term are shown in Figure 1. The biological process subgroups with the largest number of proteins were translation, followed by carbohydrate metabolic process, fatty acid biosynthetic process, proteolysis, pentose-phosphate shunt, oxidative branch, and glucose metabolic process. For molecular function, the proteins were mainly distributed in the subgroups of ATP binding, structural constituent of ribosome, and metal ion binding. Finally, the cellular component of the cytoplasm came in first, followed by the nucleus and cytosol.

The prevalent chemical reactions by which proteins were classified according to EC were Oxireductases (EC 1), transferases (EC 2), and hydrolases (EC 3) classes (Figure 2b). In the subclasses of oxidoreductases, the main groups were acting on the CH-CH group of donors (EC 1.3), acting on Ch or CH(2) groups of donors (EC 1.17), and acting on the aldehyde or oxo group of donors (EC 1.2). The most representative subclasses of transferases class included those with a transferring phosphorus-containing groups (EC 2.7) and transferring nitrogenous groups (EC 2.6). For hydrolases, those acting on carbon–nitrogen bonds (EC 3.5) and acting on acid anhydrides (EC 3.6) came first.

### 2.3. Oil Palm Metabolome under Salinity Stress

Statistical analysis on Metaboanalyst returned 2843 and 1855 peaks, respectively, in the polar-positive and polar-negative fractions (Table 1). Fifty-two peaks were differentially expressed, and eighteen were up-regulated and thirty-four were down-regulated in the polar-positive while seventy-seven were differentially expressed in the polar-negative, in which nineteen were up-regulated and fifty-eight were down-regulated.

All 129 peaks differentially expressed were then submitted to functional interpretation via analysis in the MS Peaks to Pathway module, and the combined mummichog and GSEA pathway meta-analysis resulted in a list of 19 differentially expressed metabolites (DEMs), which was then submitted to the pathway topology analysis module (Table 4). The monobactam biosynthesis (map00261); arginine biosynthesis (map00220); beta-alanine metabolism (map00410); pentose phosphate pathway (map00030); carbon fixation in photosynthetic organisms (map00710); alanine, aspartate and glutamate metabolism (map00250); galactose metabolism (map00052); and glutathione metabolism (map00480) pathways came out as the one with a raw *p* ≤ 0.05 (Figure 4).

### 2.4. Integrating Oil Palm Transcriptome, Proteome and Metabolome

A total of 510 enzymes (436 from transcriptomics analysis and 74 from proteomics analysis) (Appendix A) and 19 metabolites from metabolomics analysis (Table 3), all selected as differentially expressed in the leaves of young oil palm plants (stressed/control), were submitted to MOI analysis.

By applying the Omics Fusion platform to perform the MOI analysis, results revealed a group of eleven pathways affected by salinity stress, and with at least one molecule differentially expressed in each one of the three omics platforms used (Table 5). The Cysteine and methionine metabolism (map00270) and the Glycolysis/Gluconeogenesis (map00010) pathways came tied first in this list, each one with 20 unique molecules from the transcriptome/proteome/metabolome integrative analysis (Appendix A).

## 3. Discussion

Soil salinization reduces plant growth and productivity of most terrestrial crops with economic importance, including oil palm [13]. In the case of oil palm, Refs. [6,18,19] reported the development of salinization protocols, which are necessary to study the response of *E. guineensis* to this abiotic stress in the search for some intraspecific trait variability. Such protocols are also needed to select tolerant oil palm genotypes developed via genetic engineering or genome editing strategies. These two studies generated not only morphophysiological, biochemical, and molecular insights into the response of this species to this abiotic stress but also reported on the ionic imbalance in the substrate, roots, and leaves of young oil palm plants under salinity stress.

Salinity tolerance is a multigenic trait that governs physiological, biochemical, and molecular mechanisms to facilitate water retention and/or acquisition, protect chloroplast functions, and maintain ion homeostasis [13]. Datasets in genomics, transcriptomics, proteomics, metabolomics, epigenomics, ionomics, and phenomics are accumulating everywhere, intending to gain insights into the mechanisms behind plant interaction with abiotic stresses; however, due to the molecular complexity of such interaction, single-omics analyses (SOA) will have limited power in delivering a more systemic and accurate picture of those responses. Multi-Omics Integration (MOI) strategies [20] are a new window of opportunity facilitating hypothesis generation, leading to the non-trivial challenge of unraveling the mechanisms behind this multigenic trait.

In the present study, SOA showed that carbohydrate metabolism and translation were the most affected biological process subcategories for differentially expressed genes and proteins, respectively. In addition to being a substrate for energy production, carbohydrates play a role in plant stress perception and signal transduction and can also mediate osmotic regulation and carbohydrate distribution [21].

From the translation point of view, the protein synthesis machinery is quite sensitive to salt since the production of new proteins is crucial for salinity tolerance [22]. Salinity-tolerant species have a more efficient system for regulating transcription, synthesis, and protein processing when compared to sensitive species [23]. Genes encoding the plastid translation machinery in *Arabidopsis thaliana* are salt responsive, indicating a possible role in supporting chloroplast functionality [23]. *Reaumuria soongarica* (Pall.) Maxim., a salt-tolerant species, showed a complex pattern of protein expression, mainly those involved in translation, ribosomal structure, and biogenesis [24].

The dominant molecular function for DE enzymes identified in this study was ATP-binding proteins. These enzymes use the energy of ATP hydrolysis to catalyze a series of chemical reactions [25]. Most ATP-binding proteins are intracellular and extracellular transmembrane proteins, participating in the movement of various molecules and, under stress conditions, in intracellular osmotic balance maintenance [26]. ABC transporters that constitute one of the most populated families of proteins driven by ATP hydrolysis revealed a complex expression pattern in *E. guineensis* under drought stress, pointing to their role in controlling the influx and efflux of chemical molecules while in water scarcity [27].

Integral membrane components, the most affected subcategory of cellular components, include, in this category, proteins incorporated into cell membranes. Salt stress causes damage to the cell membrane, altering its permeability, lipid composition, and enzyme activity [28]. Several factors cause changes in the structure of cell membrane components during salt stress; among them, the excessive production of reactive oxygen species (ROS) is highlighted, which causes conformational changes in membrane proteins and lipid peroxidation, reducing the efficiency of transport systems and increasing membrane permeability [29].

In salinity-tolerant cultivars, an increase in the antioxidant defense system occurs, reducing lipid peroxidation and maintaining adequate levels of membrane permeability. On the other hand, there is an increase in the leakage of electrolytes from the membranes, which indicates a loss of membrane integrity in the sensitive plants [29]. The increase in electrolyte leakage has already been evident in oil palm leaves, indicating possible damage caused by salinity, with direct consequences in photosynthetic capacity reduction and biomass accumulation [18].

In the present study, an attempt to integrate three distinct omics platforms—transcriptomics, proteomics, and metabolomics—was reported for the first time to gain further insights into the mechanisms behind the response of young oil palm plants to salinity stress. The MOI strategy used in the present study is a pathway-based approach for integrating omics datasets. Such integration was only possible due to the selection and characterization of salt-responsive genes coding for enzymes in the oil palm reference genome. Enzymes catalyzing reactions in a metabolic pathway are the bridges to connect transcriptomics, proteomics, and metabolomics datasets in such an integratomics study.

The present MOI study revealed eleven pathways affected by the salinity stress in the leaves of the young oil palm plant, with at least one molecule differentially expressed in all three platforms used. The Cysteine and methionine metabolism (map00270) and Glycolysis/Gluconeogenesis (map00010) pathways were the most affected ones. Even though this study identified other pathways, further discussion will concentrate only on these two.

Reactions that promote cysteine (Cys) biosynthesis are involved in the pathway of cysteine and methionine metabolism [30]. Cys acts as a sulfur donor for the biosynthesis of many essential bio-molecules, such as methionine, vitamins, co-factors, and Fe-S groups, and for the production of glutathione (GSH), considered the principal determinant of cellular redox homeostasis [30]. The enzymes serine O-acetyltransferase (EC 2.3.1.30) and cysteine synthase (EC 2.5.1.47) usually carry out the Cysteine biosynthesis in two steps. These enzymes are highly conserved in plants and are responsible for maintaining homeostasis between cysteine consumption and sulfate reduction [31].

Among the enzymes integrated into the cysteine and methionine metabolism pathway, seven and twelve were up- and down-regulated in the leaves of young oil palm plants under salinity stress, respectively. Serine O-acetyltransferase experienced an approximately 11-fold increase in expression, while L-lactate dehydrogenase experienced a decrease of about 90% in its original expression level. As cysteine is the first organic compound in the primary metabolism of sulfate, the elevated transcription of serine O-acetyltransferase may indicate that sulfate entry into the pathway plays a role in the saline stress response in oil palm. In tobacco, plants over-expressing bacterial serine O-acetyltransferase conferred resistance to high levels of oxidative stress with a four-fold higher cysteine expression [32]. Recently, Ref. [33] demonstrated that the exogenous application of nitric oxide (NO), a compound that regulates the response to different stresses in plants, increased the content of enzymes synthesizing Cys, helping maintain the cellular homeostasis in plants under the osmotic tension.

Amino acid methionine has nutritional value for plants, participating in the initiation of translation, in addition to being a precursor of S-Adenosyl methionine (SAM), the donor of the methyl group that regulates different essential cellular processes, such as cell division, synthesis cell wall, chlorophyll synthesis, and membrane synthesis [34]. SAM is synthesized from adenosine triphosphate (ATP) and methionine by the enzyme S-adenosylmethionine synthetase—SAMS (EC 2.5.1.6). The present study showed that this enzyme had a 2.6 fold increase in expression under salinity stress. Overexpression of the SsSAMS2 gene from the halophyte plant of *Suaeda salsa* L. in transgenic tobacco plants enhanced salt tolerance, as indicated by maintaining a higher photosynthetic rate and accumulation of more biomass [35].

GSH is a low molecular weight thiol crucial for maintaining the regulation of cellular redox homeostasis [30]. Two ATP-dependent enzymes, glutamate-cysteine ligase (EC:6.3.2.2) and GSH synthetase (EC:6.3.2.3), catalyze GSH synthesis from cysteine, glutamate, and glycine [36]. In the present study, the metabolite glutathione (C00051) upregulated 2.9 fold while the enzyme glutathione synthase downregulated to 70% of its original levels in the leaves of young oil palm plants under salinity stress. Meanwhile, all versions of glutamate–cysteine ligase found in the reference genome of oil palm [37,38] were non-DE. The exogenous application of GSH reversed the effects of salt stress on seedlings of tomatoes, as well as the expression and activities of enzymes related to the synthesis and metabolism of GSH, including gamma-glutamylcysteine synthetase (γ-ECS) and glutathione synthetase (GS), among others [39].

The glycolysis pathway directly supplies energy to plant cells from reactions that oxidize hexoses to produce ATP and pyruvate, the latter acting as a substrate for entry into the citric acid cycle (TCA). Conversely, the gluconeogenesis pathway synthesizes hexoses using low molecular weight compounds to meet energy needs under the conditions of reduced glucose supply [40]. The ATP-dependent 6-phosphofructokinase 2 (PKF) and pyruvate kinase (PK) enzymes from the glycolytic pathway did downregulate in the leaves of young oil palm plants under saline stress. Those two enzymes, together with hexokinase, are regulators of glycolysis, as they participate in irreversible reactions [40]. The energy production via glycolysis plays a role in the saline stress response in plants as it provides ATP to support the stress condition [41].

That was evident in the study by [41], where salt stress inhibited the growth of *Cucumis sativus* L. with a significant reduction in ATP production rates [41] and applied exogenous putrescine (Put), reversing the saline stress with positive modulation in the PFK and PK levels. In halophyte species *Bruguiera sexangular*, both PFK and PK enzymes increased expression in response to long-term salinity [42]. This suggests that increased PFK and PK activity increases the activity of the glycolytic pathway to maintain normal physiological metabolism under saline stress conditions in halophyte species. Salinity stress possibly promoted a reduction in ATP production as it negatively affected enzymes in the flow of glycolysis and TCA in oil palm.

Fructose-1,6-bisphosphatase (FBPase) and phosphate dikinase (PPDK), enzymes of the gluconeogenesis pathway, were negatively regulated in oil palm under salt stress. Under saline stress conditions, the active synthesis of sugars by this route contributes to mitigating the osmotic stress effect resulting from the submission of plants to a saline environment. In maize (*Zea mays* L.), the photosynthesis rate was similar between control plants and plants under neutral salt stress, suggesting that gluconeogenesis acted on the active synthesis of sugars and the maintenance of osmotic balance [43].

The overexpression of TaFBA-10 in *A. thaliana* (L.) Heynh increased FAB activity with positive effects on scavenging ROS under cold stress, whereas chlorophyll content was severely affected [44]. The PPDK enzyme, in turn, is an enzyme involved in the regulation of the C4 pathway in plants. PPDK enzyme activity increases in salinity tolerant accessions of *Miscanthus sinensis* Andersson [45]. In this manner, the activity of this enzyme compensates for the suppression of the Calvin Cycle by saline stress.

Another enzyme that participates in energy production is L-lactate dehydrogenase type B (LDH). This enzyme did downregulate in young oil palm plants under salinity stress. Lactate dehydrogenase (LDH) converts pyruvate to lactate that regenerates NAD + to maintain cellular respiration under anaerobic conditions. Under flood stress conditions, the initiation of fermentation responds to keeping energy supply in hypoxia conditions [46]. The downregulation of the LDH enzyme in palm oil at 12 DAT indicates a deficiency in response to salinity stress and may indicate a possible anaerobic condition caused by it.

Vieira and colleagues showed that young oil palm plants are sensitive to high concentrations of NaCl [6]. The present MOI study, which used datasets generated from leaf tissue collected by Vieira and colleagues, showed that enzymes competing for energy production in the glycolysis and gluconeogenesis pathways were negatively affected by salinity stress in the leaves of young oil palm plants. Concomitantly, gluconeogenesis, which involves the synthesis of glucose from non-carbohydrate substances, apparently does not represent an immediate response to reduced glucose supply in this oilseed crop under such stress.

The samples for transcriptome, metabolome, and proteome analyses were collected at once, exactly 12 days after the onset of the stress, using a split sample study design. In terms of data integration, accordingly to [47], the ideal situation is to have samples originating from the same biological source material and obtained at the same time—a piece of tissue may be cut into several sections and one used for a specific omics platform analysis, whilst the other is used to another one. In such design, the samples are more similar in that they all are assumed to produce data without batch effects between the different omics data sets [47].

It is clear that the small number of DE metabolites was the main limitation of the pathway-based MOI approach used in this study. The biggest number of DE metabolites in the eleven most affected pathways identified in the MOI analysis was three, while there were up to six proteins and 17 transcripts. The possible main reason to the fact that only 19 metabolites were differentially expressed in the leaves of young oil palm plants submitted to salinity stress was the mmetabolomics approach used. The untargeted metabolomics is an exciting technology for searching for novel metabolic perturbations in various biological systems, allowing the profile of many hundreds or thousands peaks with varying chemical properties at once; however, there are still various obstacles, such as the limited capability to identify novel compounds of interest and the need for advanced and more robust databases [48]. In the present study, we used the latest KEGG version of the *O. sativa* pathway library.

## 4. Materials and Methods

### 4.1. Plant Material, Experimental Design and Saline Stress

The oil palm plants used in this study were clones regenerated out of embryogenic calluses obtained from the leaves of an adult plant—genotype AM33, a Deli × Ghana from ASD Costa Rica, as previously reported by [6]. Before starting the experiments, plants were standardized accordingly to the developmental stage, size, and number of leaves. They were in the growth stage known as bifid saplings, and the experiment was performed in March 2018 in a greenhouse at Embrapa Agroenergy in Brasília, DF, Brazil (S-15.732°, W-47.900°). The main environmental variables (temperature, humidity, and radiation) fluctuated according to the weather conditions and underwent monitoring throughout the entire experimental period using the data collected at a nearby meteorological station (S-15.789°, W-47.925°).

The experiment consisted of five salinity levels (0.0, 0.5, 1.0, 1.5, and 2.0 g of NaCl per 100 g of substrate (a mixture of vermiculite, soil, and the Bioplant commercial substrate (Bioplant Agrícola Ltd.a., Nova Ponte, MG, Brazil), in a 1:1:1 ratio, on a dry basis), with four replicates in a completely randomized design (for additional details, see [6]). The substrate mixture was fertilized using 2.5 g L^−1^ of the N-P2O5-K2O formula (20-20-20). For the omics (transcriptomics, metabolomics, and proteomics) analysis described in the present study, we collected the apical leaves from control and stressed plants (0.0 and 2.0 g of NaCl per 100 g of substrate) 12 days after imposition of the treatments (DAT).

### 4.2. Transcriptomics Data Analysis

Leaves harvested from control and stressed plants were immediately immersed in liquid nitrogen and stored at −80 °C until RNA extraction; three plants for treatments. Details regarding total RNA extraction and quality analysis, library preparation, and sequencing are in [15,19]. RNA-Seq raw sequence data are in the Sequence Read Archive (SRA) database of the National Center for Biotechnology Information (NCBI) under BioProject number PRJNA573093.

All the transcriptomics analysis was performed with OmicsBox platform—version 2.0.36 [49], as previously described by [15,17]. The oil palm genome [19,20]—downloaded from NCBI (BioProject PRJNA268357; BioSample SAMN02981535) in September 2021—was the reference genome for RNA-Seq data alignment. The pairwise differential expression analysis between experimental conditions (Stressed Plants—12 DAT X Control—12 DAT) was performed through edgeR software version 3.28.0 [50], applying a simple design and an exact statistical test without the use of a filter for low counts genes.

### 4.3. Proteomics Data Analysis

Leaves samples for proteomics analysis were harvested, immediately immersed in liquid nitrogen, and then stored at −80 °C until protein extraction; three plants for control and three from stressed plants. Approximately 5.0 g of ground tissue—with 0.02 g/g of PVP (polyvinylpolypyrrolidone) added to it—was weighed and mixed with 3.0 mL of buffer (50 mM Tris HCl + 14 mM β-mercaptoethanol, pH 7.5) and 30 µL of protease inhibitor. After gently stirring the suspension on ice for 10 min, it was centrifuged at 10.000 G at 4.0 °C for 15 min. Then, 1.0 mL of the supernatant was transferred to 2.0 mL microtubes, mixed with 1.0 mL of 10% TCA (trichloroacetic acid) solution in acetone, kept at −20 °C for 2 h for protein precipitation, and then centrifuged at 10,000 G at 4.0 °C for 15 min. The protein pellet underwent wash with ice-cold 80% acetone, followed by centrifugation under the same conditions as above. After washing twice, we stored it at −80 °C until protein quantification [51] and visualization in an SDS-PAGE Gel.

After protein quantification, all samples went to the GenOne company (Rio de Janeiro, RJ, Brazil) fort protein preparation and LC-MS/MS analysis. After undergoing treatment with 10 mM DTT at 56 °C for 30 min, followed by 40 mM iodoacetamide (IDA) at room temperature in the dark and also for 30 min. Then, samples were incubated for 20 h at 37 °C with trypsin (1:50) in a thermomixer at 800 rpm. At last, after adding 50 µL of 95% acetonitrile and 5% TFA, samples were stirred three times at 1000 rpm for 15 min for tryptic peptides extraction, vacuum dried, and dissolved in 20 μL of 0.1% formic acid in water.

For a global proteomics analysis, we adopted a label-free quantitation approach using spectral counting by LC-MS/MS passing the samples through a nano-high performance liquid chromatography (EASY 1000; Thermo Fisher, Waltham, MA, USA) coupled to Orbitrap Q Exactive Plus (Thermo Scientific, Waltham, MA, USA) mass spectrometer. An MS scan spectra ranging from 375 to 2000 *m*/*z* were acquired using a resolution of 70,000 in the Orbitrap. We used the Xcalibur software (version 2.0.7)(Thermo Scientific, Waltham, MA, USA) to obtain the data in biological triplicates.

The PatternLab for Proteomics V software [23] was used to process the raw files. We used the Comet algorithm [52], the *E. guineensis* Uniprot reference database (30,667 entries), and 123 common contaminant proteins (Proteome ID: UP000504607) to perform peptide sequence matching (PSM) and employed a target-reverse strategy to increase confidence in protein identifications [53]. The search considered semi-specific candidates and allowed a maximum of two missed cleavages. Fixed cysteine carbamidomethylation and variable methionine oxidation were applied; the Comet search engine used a precursor mass tolerance of 40 ppm and a fragment compartment tolerance of 0.02.

We employed the SEPro—Search Engine Processor—module of PatternLab [54] to validate the peptide spectrum matches and, subsequently, grouped identifications by enzymatic specificity (semi-specific), resulting in two distinct subgroups. Then, we applied XCorr, DeltaCN, Spectral Counting Score, and Peaks Matched values to generate a Bayesian discriminator. SEPro automatically establishes a cutoff score to accept a 1% false discovery rate (FDR) based on the number of decoys performed independently on each subset of data, resulting in a false positive rate independent of the triptych status. We chose a minimum sequence length of six amino acid residues and discarded similar proteins that represent an identical sequence and consist of a fragment of another one. At last, a final list of mapped proteins was composed only of PSMs with less than five ppm.

### 4.4. Metabolomics Data Analysis

Leaves harvested from control and stressed plants were immediately immersed in liquid nitrogen and stored at −80 °C until metabolite extraction: four plants for treatments. Before solvent extraction, all samples underwent grounding in liquid nitrogen. The solvents used were methanol grade UHPLC, acetonitrile grade LC-MS, formic acid grade LC-MS, sodium hydroxide ACS grade LC-MS, all from Sigma-Aldrich, and water treated in a Milli-Q system from Millipore. We employed a well-established protocol [16,55,56] to extract the metabolites in three phases (polar, non-polar, and protein pellet). Aliquots of 50 mg of ground sample were transferred to 2 mL microtubes, and then 1 mL of a mixture of 1:3 (*v*:*v*) methanol/methyl tert-butyl ether (MTBE) at −20 °C was added. Homogenization on an orbital shaker at 4.0 °C and ultrasound treatment in an ice bath were each performed for 10 min. As the next step, we added 500 μL of a mixture of 1:3 (*v:v*) methanol/water to each microtube. After centrifugation (15,300× *g* at 4.0 °C for 5 min), an upper non-polar (green) and a lower polar (brown) phase and a protein pellet remained in each microtube. After transferring both fractions separately to 1.5 mL microtubes, they were submitted to a Speed vac system (Centrivap, Labconco) to be vacuum dried. Finally, the dry-fraction, resuspended in 500 μL of 1:3 (*v:v*) methanol and water mixture and transferred to vials, were now ready for analysis.

Analytical method UHPLC–MS/MS (ultra-high performance liquid chromatography and tandem mass spectrometry) was used in this study. The UHPLC system (Nexera X2, Shimadzu Corporation, Kyoto, Japan) was equipped with a reverse-phase column from Waters Technologies (Acquity UPLC HSS T3, 1.8 μm, 2.1 by 150 mm at 35 °C). Solvent A was 0.1% (*v:v*) formic acid in water and solvent B was 0.1% (*v:v*) formic acid in acetonitrile/methanol (70:30, *v:v*). The gradient elution used, with a flow rate of 0.4 mL min^−1^, was as follows: 0–1 min isocratic, 0% B; 1–3 min, 5% B; 3–10 min, 50% B; 10–13 min, 100% B; 13–15 min isocratic, 100% B; then, 5 min rebalancing was conducted to the initial conditions. The rate of acquisition spectra was 3.00 Hz, mass range *m/z* 70–1200 for the polar fraction analysis and *m/z* 300–1600 for the lipidic fraction.

High-resolution mass spectrometry was used for detection (MaXis 4G Q-TOF MS, Bruker Daltonics) equipped with an electrospray source in positive (ESI-(+)-MS) and negative (ESI-(−)-MS) modes. The settings of the mass spectrometer were as follows: capillary voltage, 3800 V; dry gas flow, 9 L min^−1^; dry temperature, 200 °C; nebulizer pressure, 4 bar; final plate offset, 500 V. For the external calibration of the equipment, we used a sodium formate solution (10 mM HCOONa solution in 50:50 *v:v* isopropanol and water containing 0.2% formic acid) injected through a six-way valve at the beginning of each chromatographic run. Ampicillin ([M+H] + *m/z* 350.1186729 and [M-H]- *m/z* 348.1028826) was the internal standard for later peak normalization on data analysis.

DataAnalysis 4.2 software (Bruker Daltonics, Bremen, Germany) was the first used to analyze the raw data from UHPLC-MS, as mzMXL files. Pre-processing of data was performed using XCMS Online [57,58], including peak detection, retention time correction, and alignment of the metabolites. CentWave was used for peak detection (Δ*m/z* = 10 ppm; minimum peak width, 5 s; maximum peak width, 20 s). For the alignment of retention times, the parameters were as follows: mzwid = 0.015; minfrac = 0.5; bw = 5. The unpaired parametric *t*-test (Welch t-test) was used for the statistical analysis at the pre-processing stage. Then, a data set was created from control (0.0 g) and stressed plants subjected to NaCl/100 g of the fresh substrate at 12 DAT. All with four biological repeats.

The pre-processed data (csv file) underwent analysis in the Statistical Analysis module of the MetaboAnalyst 5.0 [59,60]. The scaling used was the Pareto method [61]. Afterward, the differentially expressed peaks (DEPs) selected were those passing the criteria of false rate discovery (FDR) ≤ 0.05 and Log_2_ (fold change (FC)) ≠ 1. When using the MS Peaks to Pathway module to analyze the selected DEPs, we employed the following parameters: molecular weight tolerance of 5 ppm; mixed ion mode; joint analysis using both the mummichog [62] and Gene Set Enrichment Analysis—GSEA [63] algorithms; the latest KEGG version of the *O. sativa* pathway library. The *p*-value cutoff from the mummichog algorithm was at 1.0 × 10^−5^.

When two or more matched forms were observed as DEP (in the case of isotopes), the mass error was the criteria for the feature selection for the comparison with metabolite databases, keeping the smallest [56]. The mass error was also the criteria in the case of a single matched compound relative to two or more DEPs. The mass spectra of all DEPs underwent analysis for more information about the adduct forms obtained from the database comparison. Subsequently, we performed the putative annotation of the metabolites of interest by applying the filtered exact mass data to the molecular formula from KEGG.

Finally, the KEGG IDs of the matched compounds were submitted to the pathway analysis module for visualization through integrating enrichment and pathway topology analysis [64]. Parameter sets were as follows: the hypergeometric test and the latest KEGG version of the *O. sativa* pathway library.

### 4.5. Functional Annotation and Itegratomics Analysis

The results obtained using OmicsBox and PatternLab V underwent a functional classification. Distinct multiFASTA files generated were submitted to the functional classification in the BlastKOALA platform [64].

The approach used to integrate the three omics was pathway mapping, and the analysis was performed using the Omics Fusion platform [65]. Previously to the integration of multi-omics data, the NCBI accession of transcripts related to enzymes was converted to UniProt ID. Thus, the input data used were the IDs of each omics, which include UniProt Accession for transcriptomics and proteomics, and KEGG ID for metabolomics. Firstly, the data were enriched through several databases (EMBL, KEGG, NCBI, and UniProt), and then the module “KEGG feature distribution” was used to map these omics data in known pathways.

## 5. Conclusions

Previously, in addition to showing that young oil palm submitted to a high concentration of NaCl reduces the rates of CO_2_ assimilation, stomatal conductance to water vapor, and transpiration, our group also confirmed a preponderant role of transcription factors in the early response of oil palm plants to salinity stress [6,15]. Data from ionomics, phenomics, and transcriptomics (mRNA and miRNA) were employed to show that. Currently, two new omics platforms joined this list—metabolomics and proteomics—and a first MOI study was performed. For phenomics–morphophysiological characterization, data came from two salinity stress experiments carried out in November 2017 and March 2018, and all the transcriptome, metabolome, and proteome data came from one of the experiments at once, 12 days after the onset of the stress, using a split-sample study design. Extensive leaf necrosis was already visible when the samples from the stressed treatment (electrical conductivity of ~40 dS m^−1^) were collected, and one must consider that when analyzing these omics data sets.

The SOA and MOI studies here reported generated new insights on the response the early response of oil palm plants to salinity stress, pointing out genes, proteins, metabolites, and pathways directly affected by this stress. The eleven pathways identified by MOI analysis definitely appear at the top of the list as priorities for further studies. However, it is clear that two factors limited the accomplishments of the MOI study—the small number of differentially expressed metabolites identified via an untargeted metabolomics approach and the lack of data regarding the Log_2_(FC) from the proteins found exclusively in the control and stressed treatments when using the global proteomics analysis. No Log_2_(FC) from most of the DE proteins was identified, and only 19 DE metabolites limited the use of correlation studies.

## Figures and Tables

**Figure 1 plants-11-01755-f001:**
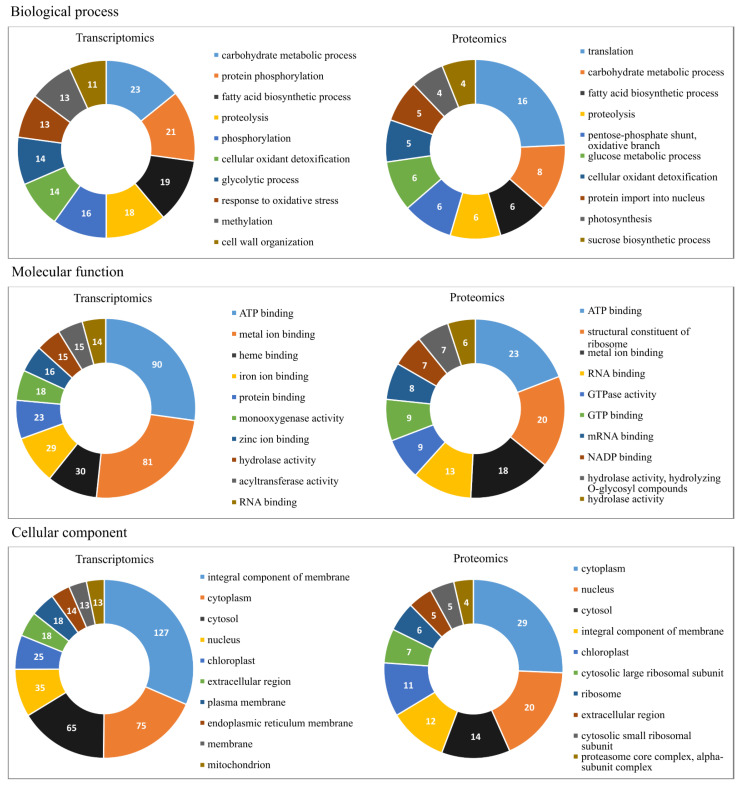
Gene Ontology (GO) annotation classification statistics graph from full-length transcriptome and proteome in the leaves of young oil palm plants under salinity stress; classified accordingly to biological process, cellular component, and molecular function. Only the ten most populated groups per GO term are shown. Numbers represent the amount of positive hits.

**Figure 2 plants-11-01755-f002:**
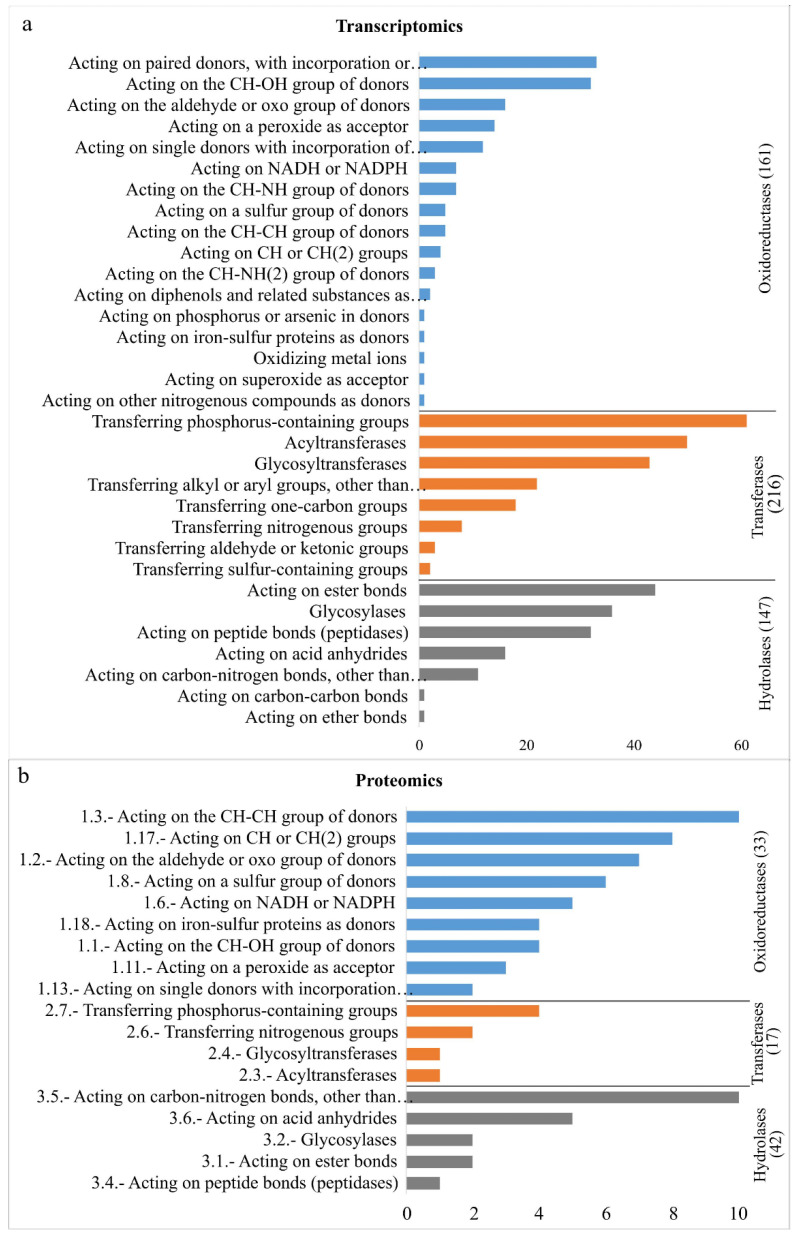
Gene Ontology (GO) annotation classification statistics graph from full-length transcriptome and proteome in the leaves of young oil palm plants under salinity stress; classified accordingly to chemical reactions by which proteins are classified according to E.C. Only the three prevalent classes are shown: oxireductases (EC 1), transferases (EC 2), and hydrolases (EC 3). (**a**)—Transcriptomics Single Analysis, and (**b**)—Proteomics Single Analysis.

**Figure 3 plants-11-01755-f003:**
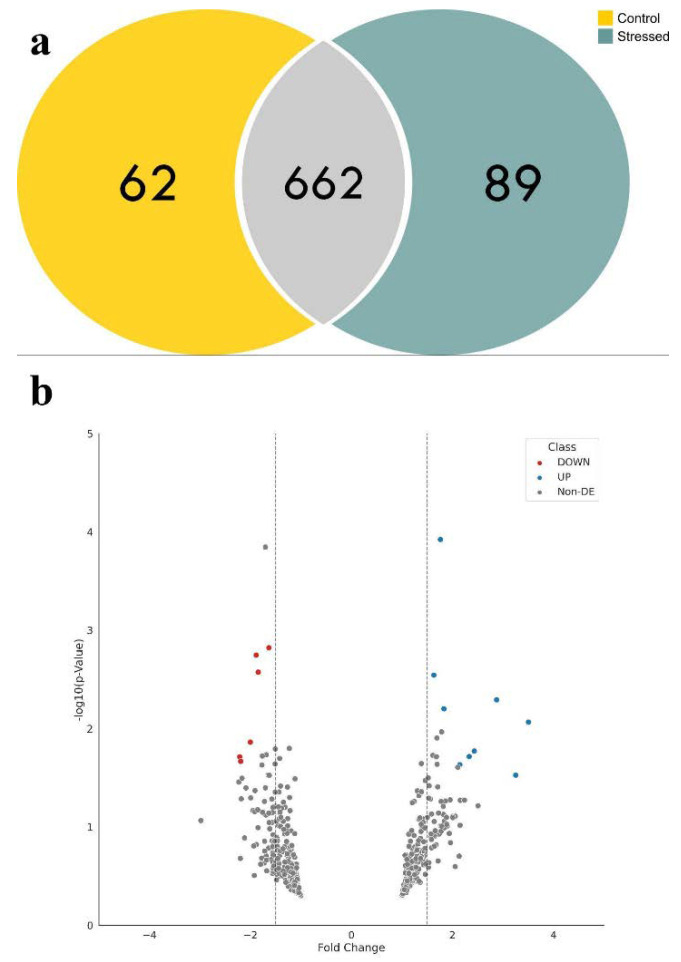
Summary of the proteomics analysis performed on the leaves of young oil palm plants under salinity stress using the PatternLab for Proteomics V software. (**a**) Control and stressed conditions shared 662 protein identifications; 62 and 89 proteins were uniquely detected in control and stressed samples, respectively; (**b**) volcano plot of the differentially abundant proteins reported by Pattern Lab’s T Fold module, where 20 proteins showed statistically significant differences in their abundance—proteins in blue were significantly up-regulated while the ones in red were significantly down-regulated between stressed and control samples.

**Figure 4 plants-11-01755-f004:**
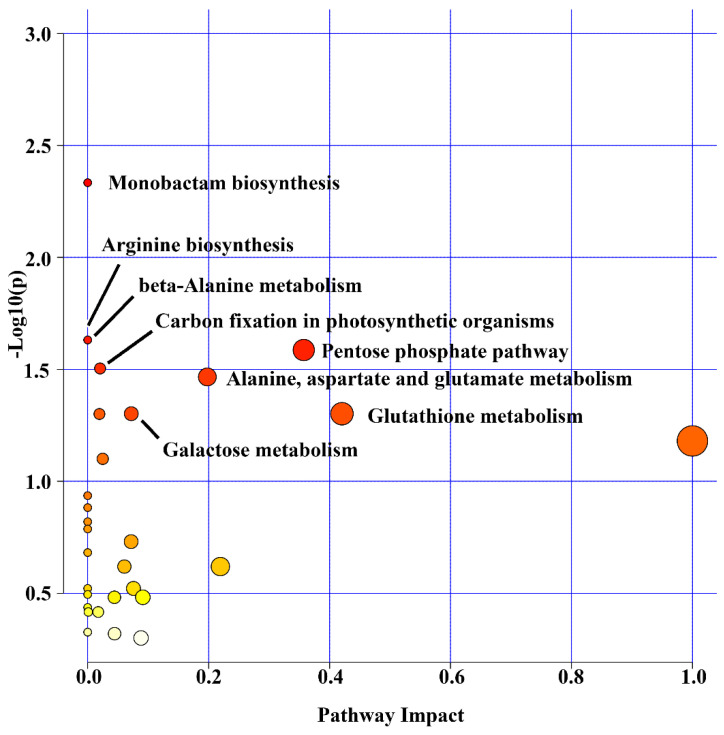
Summary of the pathway analysis in the leaves of young oil palm plants under salinity stress using the Pathway Topology Analysis modules of MetaboAnalyst 5.0. The metabolome view resulted from the analysis in the Pathway Topology Analysis module using the Hypergeometric test, the relative betweenness centrality node importance measure, and the latest KEGG version of the *Oryza sativa* pathway library. Pathway impact takes into account both node centrality parameters—betweenness centrality and degree centrality—and represents the importance of annotated compounds in a specific pathway.

**Table 1 plants-11-01755-t001:** Differentially expressed (DE) peaks and features in the leaves of young oil palm plants submitted to salinity stress selected by means of three distinct omics platforms (transcriptomics, metabolomics, and proteomics).

**Transcriptomics**	**Number of Features**	**Up**	**Down**	**Non-DE**
WGS–Proteins	43,551	1138	1590	40,823
**Metabolomics**	**Number of Peaks**	**Up**	**Down**	**Non-DE**
Positive Polar	2843	18	34	2791
Negative Polar	1855	19	58	1778
**Proteomics ***	**Number of Features**	**Up**	**Down**	**Non-DE**
LC/MS	813	101	70	642

* Up = Proteins found exclusively in stressed samples + Proteins that attended to statistical criteria of PatternLab V software; Down = Proteins found exclusively in control samples + Proteins that attended to statistical criteria of PatternLab V software [17].

**Table 2 plants-11-01755-t002:** Absolute numbers of all peptides and proteins identified via proteomics analysis in the leaves of young oil palm plants submitted to salinity stress.

	Control	Stressed	Total
Peptide Spectrum Match (PSM)	5419	5391	10,808
Total number of peptides	3234	2872	4254
Number of unique peptides	1805	1606	2426
Total number of proteins entries	1497	1436	1809
Total number of proteins using the maximum parsimony criterion	826	831	1019

**Table 3 plants-11-01755-t003:** List of the differentially expressed proteins detected in both biological conditions (Stressed and Control) with statistical significance (FDR ≤ 0.05).

Entry	Class	Fold Change	*p*-Value	Signal in Control	Signal in Stressed	Gene ID at NCBI	Description
A0A6I9RY35	UP	3.50631	0.00860	0.00027	0.00094	LOC105054572	probable inactive purple acid phosphatase 29
A0A6I9QVF6	UP	3.25426	0.02982	0.00104	0.00340	LOC105040203	GTP-binding nuclear protein
A0A6I9R375	UP	3.25426	0.02982	0.00085	0.00275	LOC105043116	GTP-binding nuclear protein
A0A6I9RFH3	UP	3.25426	0.02982	0.00104	0.00340	LOC105047773	GTP-binding nuclear protein
A0A6I9QCS1	UP	2.87620	0.00511	0.00059	0.00169	LOC105033701	Proteasome subunit alpha type
A0A6I9QQJ4	UP	2.43453	0.01697	0.00103	0.00251	LOC105039272	60S ribosomal protein L35a-1
B3TLX9	UP	2.43453	0.01697	0.00103	0.00251	LOC105037063	60S ribosomal protein L35a-1
A0A6I9QWA8	UP	2.33349	0.01927	0.00071	0.00165	LOC105039716	Succinate-semialdehyde dehydrogenase
A0A6I9RG83	UP	2.14817	0.02330	0.00062	0.00133	LOC105045986	uncharacterized protein LOC105045986
B3TLY5	UP	1.83395	0.00630	0.00105	0.00193	CAT2	Catalase
A0A6I9QQQ6	UP	1.76320	0.00012	0.00068	0.00119	LOC105039332	V-ATPase 69 kDa subunit
A0A6I9R4U7	UP	1.63284	0.00286	0.00276	0.00450	LOC105044322	Malate dehydrogenase
A0A6I9S1Z5	DOWN	−1.63290	0.00151	0.00166	0.00101	LOC105055575	ruBisCO large subunit-binding protein subunit alpha
A0A6I9QJN4	DOWN	−1.84374	0.00267	0.00177	0.00096	LOC105036569	CBBY-like protein
A0A6I9RPV6	DOWN	−1.84374	0.00267	0.00177	0.00096	LOC105051320	CBBY-like protein
A0A6J0PH47	DOWN	−1.88477	0.00179	0.00395	0.00210	LOC105044080	Ferredoxin—NADP reductase, chloroplastic
A0A6I9S9I9	DOWN	−2.00037	0.01375	0.00091	0.00045	LOC105058225	uncharacterized protein LOC105058225
A0A6I9RWU5	DOWN	−2.19127	0.02157	0.00284	0.00129	LOC105054048	actin-101
A0A6I9RC26	DOWN	−2.21145	0.01945	0.00172	0.00078	LOC105047077	sorbitol dehydrogenase isoform X2
A0A6I9RDE7	DOWN	−2.21145	0.01945	0.00172	0.00078	LOC105047077	sorbitol dehydrogenase isoform X1

**Table 4 plants-11-01755-t004:** List of metabolites identified in the leaves of young oil palm plants submitted to salinity stress via metabolomics analysis, after submitting the differentially expressed (DE) peaks to the pathway topology analysis module in MetaboAnalyst 5.0. FDR: False Discovery Rate; and FC: Fold Change.

Query Mass	Matched Compound	Matched Form	Mass Difference	Compound Name	FDR	Log_2_(FC)
145.01452	C00026	M-H[–]	2.69 × 10^−4^	Oxoglutaric acid	0.0106	–0.4146
616.17640	C00032	M[1+]	8.96 × 10^−4^	Heme	0.0039	2.8661
106.04953	C00049	M-CO+H[1+]	2.53 × 10^−4^	L-Aspartic acid	0.0292	0.9617
306.07651	C00051	M-H[–]	2.27 × 10^−5^	Glutathione	0.0204	1.5265
289.03241	C00117	M+CH3COO[–]	3.46 × 10^−5^	D-Ribose 5-phosphate	0.0475	–0.9714
427.01748	C00224	M(C13)-H[–]	1.46 × 10^−3^	Adenosine phosphosulfate	0.0172	–0.6544
172.98600	C00262	M+K-2H[–]	7.55 × 10^−4^	Hypoxanthine	0.0004	–1.5351
203.22237	C00750	M+H[1+]	6.58 × 10^−4^	Spermine	0.0036	2.4559
163.04033	C00811	M-H[–]	2.65 × 10^−4^	4-Hydroxycinnamic acid	0.0065	–0.3818
162.02134	C01419	M-NH3+H[1+]	6.49 × 10^−4^	Cysteinylglycine	0.0263	1.2145
260.02535	C05345	M(C13)-H[–]	4.92 × 10^−4^	Beta-D-Fructose 6-phosphate	0.0489	–1.0337
359.11946	C05399	M-H+O[–]	3.09 × 10^−5^	Melibiitol	0.0103	–1.6922
254.09610	C05401	M(C13)-H[–]	1.95 × 10^−4^	Galactosylglycerol	0.0410	–0.7515
326.09623	C05839	M(C13)-H[–]	6.61 × 10^−5^	cis-beta-D-Glucosyl-2-hydroxycinnamate	0.0472	–1.4286
277.06946	C05911	M-CO+H[1+]	1.11 × 10^−3^	Pentahydroxyflavanone	0.0143	–1.0759
337.05555	C10107	M+H2O+H[1+]	1.09 × 10^−4^	Myricetin	0.0313	–2.4012
337.00976	C11453	M+CH3COO[–]	8.02 × 10^−4^	2-C-Methyl-D-erythritol 2,4-cyclodiphosphate	0.0272	0.8232
259.02223	C17214	M+Cl37[–]	1.45 × 10^−4^	2-(3′-Methylthio)propylmalic acid	0.0222	–0.9440
447.91027	G00005	M(C13)+2H [2+]	1.30 × 10^−3^	(GlcNAc)2 (Man)3 (PP-Dol)1	0.0263	0.4044

**Table 5 plants-11-01755-t005:** List of top eleven pathways affected by salinity stress obtained via Multi-Omics Integration (MOI). Transcriptomics, proteomics, and metabolomics data from leaves of young oil palm plants after being under 0.0 (control) and 2.0 (stressed) g of NaCl/100 g of substrate for 12 days.

Pathway	Pathway ID	Occurrence of Transcripts	Occurrence of Proteins	Occurrence of Metabolites	Occurrence of Unique Molecule
Cysteine and methionine metabolism	270	15	5	2	20
Glycolysis/Gluconeogenesis	10	17	3	1	20
Glyoxylate and dicarboxylate metabolism	630	14	4	1	16
Carbon fixation in photosynthetic organisms	710	12	2	2	15
Glycine, serine and threonine metabolism	260	11	2	1	14
Pentose phosphate pathway	30	10	4	2	14
Glutathione metabolism	480	9	3	3	13
Amino sugar and nucleotide sugar metabolism	520	10	2	1	12
Carbon fixation pathways in prokaryotes	720	7	6	1	11
Citrate cycle (TCA cycle)	20	5	4	1	8
Butanoate metabolism	650	4	2	1	7

## Data Availability

The data sets used and/or analyzed during the current study are available from the corresponding author upon reasonable request.

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
