# Peer review of "Insights from a Multi-Omics Integration (MOI) Study in Oil Palm (Elaeis guineensis Jacq.) Response to Abiotic Stresses: Part One—Salinity"

_plants, 2022, doi:10.3390/plants11131755_

Round 1

Reviewer 1 Report

The MS entiled "Insights from a Multi-Omics Integration (MOI) study in oil palm (Elaeis guineensis Jacq.) response to abiotic stresses: Part one Salinity is well written and may be accepted for publications after the below revisions.

P.1-L.14-15 please re-phrase the sentence.

Introduction:

What are the translational value of omics, please add at least  para on this and correlate on salinity. Why omics important for salinity?

Please provide the pathway ( KEGG).

How much do metabolites or genes downregulate and upregulate? Please provide the table in the MS. Kindly add in  Abstract section and also in results.

I could not see the figs in the MS.

 P.4  L. 128-129: 62 and 89 proteins were uniquely detected in control and stressed samples, respectively (Figure 3a, Supplementary Table S3).

Kindly  provide the important "20" stressed proteins in the table on the basis of fold changes and p value. The remaining can be included in the supplementary file.

In methods:

In 4.1 W-47.925°)..    delete  ………………(.)

Author Response

Response to Reviewers

Reviewer 01.

The MS entiled "Insights from a Multi-Omics Integration (MOI) study in oil palm (Elaeis guineensis Jacq.) response to abiotic stresses: Part one – Salinity is well written and may be accepted for publications after the below revisions.

P.1-L.14-15 please re-phrase the sentence.

Response: Thank you for your comment. Done! Please, see revised version of the MS.

Introduction:

What are the translational value of omics, please add at least  para on this and correlate on salinity. Why omics important for salinity?

Response: Thank you for your comment. The use of the different omics platforms available nowadays, alone or via an integrative strategy, is allowing researchers everywhere to gain many more insights in the response of a plant to an environmental stress than before the "omics era". The MOI strategy - something that we can say is still in its early stage - will definitely facilitate the hypothesis generation leading to an even deeper understanding of the functions and mechanisms behind the plant x environment interactions. Salinity is just one of the many environmental stresses, which understanding is gaining so much from this new era. We see what you want to achieve when you ask us to add a paragraph about the translational value of omics, correlating with salinity. However, we decided not to do so here in this MS. As you can see from the title of this MS, this is only the first chapter we wrote applying MOI to study the response of oil palm to abiotic stresses. As second chapter (Part two - Drought) is already been written, and there we believe is the best moment to stress the translational value of omics. Based on that, I kindly ask you to accept our response as it is now to your first comment. All other comments and questions you presented were accepted and resulted in important modifications in the MS (Please, see the revised version of the MS).

Please provide the pathway (KEGG).

Response: Thank you for your comment. We (the authors) accepted your suggestion and added the KEGG numbers for each one of the pathways mentioned (Please, see the revised version of the MS).

How much do metabolites or genes downregulate and upregulate? Please provide the table in the MS. Kindly add in Abstract section and also in results.

Response: Thank you for your question. We accepted your suggestion and added two new columns to Table 3, where you are going to find the False Discovery Rate (FDR) and the Log2(FC) from the differentially expressed metabolites. Regarding the genes/proteins differentially expressed in the transcriptomics part of the study, they are provided in the Supplemental Table S1. Besides, we also added this information in the Abstract and Results sections (Please, see the revised version of the MS).

I could not see the figs in the MS.

Response: Thank you for your comment. We the authors thought that the reviewers would have access to the figures, tables, and supplementary material submitted together with the main document; but apparently that was not the case. So, we did add all figures and tables to the revised version of the MS. We hope that this will help you in your evaluation of this MS.

 P.4  L. 128-129: 62 and 89 proteins were uniquely detected in control and stressed samples, respectively (Figure 3a, Supplementary Table S3).

Kindly  provide the important "20" stressed proteins in the table on the basis of fold changes and p value. The remaining can be included in the supplementary file.

Response: Thank you for your comment. A new table was added to the MS (Table 3 in the revised version of the MS). This table shows the 20 differentially expressed proteins detected in both biological conditions (Stressed and Control) with statistical significance (FDR ≤ 0.05). Please, see the revised version of the MS.

In methods:

In 4.1 W-47.925°)..    delete  ………………(.)

Response: Thank you for your comment. Done! Please, see revised version of the MS.

Reviewer 2 Report

In the material and method section authors have used 0.0 to 2 g of NaCl per 100 g of substrate)

The questions are that:

1-     What do they mean by substrate? It must be explained more clearly and in details

2-     On what basis do the authors claim that these concentrations are very high salinity stress for this plant?

Tables and figures are missing in the manuscript and without them it is not possible to make a correct judgment. Authors should include relevant tables and figures in a comprehensible way for readers in their manuscript. Therefore, the whole manuscript are descriptive results. It is understandable that how SOA, and MOI analysis of the metabolome, transcriptome, and proteome profiles show differences among salinity levels.

It is not correct to cite the references in the result section. The authors must cite the reference in the discussion section. For example, see page 3, line 85:

The results are written in the style of materials and methods in some cases. There is so many of these errors in this manuscript. For example: in page 3 line 89, "This group of 2,728 DE proteins sequences were submitted to analysisinthe90 BlastKOALA annotation tool for K number assignment". In the results section, authors only need to write the results, or: "A global proteomics analysis was performed by a label-free quantitation approachusing118 spectral counting generated by LC-MS/MS. Proteins were identified and quantifiedusing119 PatternLab V following the criteria mentioned in the Materials & Methods section". These sentences must be deleted or included in material and methods

Therefore, I regret to say this manuscript in this form is not suitable for publication in Plants unless there are fundamental changes in it.

Author Response

Response to Reviewer 02

In the material and method section authors have used 0.0 to 2 g of NaCl per 100 g of substrate)

The questions are that:

1- What do they mean by substrate? It must be explained more clearly and in details.

Response: Thank you for your question. We (the authors) modified the text to explain more clearly and in details, as requested. The new text in the revised version of the MS is: "The experiment consisted of five salinity levels (0.0, 0.5, 1.0, 1.5, and 2.0 g of NaCl per 100 g of substrate [a mixture of vermiculite, soil, and the Bioplant commercial substrate (Bioplant Agrícola Ltda., Nova Ponte, MG, Brazil), in a 1:1:1 ratio, on a dry basis], with four replicates in a completely randomized design (for additional details, see: [6]). The substrate mixture was fertilized using 2.5 g L-1 of the N-P2O5-K2O formula (20-20-20). For the omics (transcriptomics, metabolomics, and proteomics) analysis described in the present study, we collected the apical leaves from control and stressed plants (0.0 and 2.0 g of NaCl per 100 g of substrate) 12 days after imposition of the treatments (DAT)."

2- On what basis do the authors claim that these concentrations are very high salinity stress for this plant?

Response: Thank you for your question. The study described in this MS used transcriptomics, metabolomics, and proteomics data collected from leaves obtained in a previous study (Vieira et al., 2020; or [6]). In that previous study, we did measure the electrical conductivity (EC) for all treatments tested (see Figure 2 in Vieira et al., 2020). In the stress treatment (2.0 g of NaCl per 100 g of substrate), the EC is about 40 dS m-1. This level of salinity is then 10 times higher than the minimal level in a soil for it to be considered saline (EC > 4 dS m-1). So, this level of salinity is very high not only to oil palm (Elaeis guineensis), but also to any terrestrial plant species.

To help on linking these two studies, Vieira et al. (2020) and the MS, we modified the last paragraph in the introduction section. Please, see the revised version of the MS.

Tables and figures are missing in the manuscript and without them it is not possible to make a correct judgment. Authors should include relevant tables and figures in a comprehensible way for readers in their manuscript. Therefore, the whole manuscript are descriptive results. It is understandable that how SOA, and MOI analysis of the metabolome, transcriptome, and proteome profiles show differences among salinity levels.

Response: Thank you for your comment. We thought that the reviewers would have access to the figures, tables, and supplementary material submitted together with the main document; but apparently that was not the case. So, we did add all figures and tables to the revised version of the MS. We hope that this will help you in your evaluation of this MS.

It is not correct to cite the references in the result section. The authors must cite the reference in the discussion section. For example, see page 3, line 85:

The results are written in the style of materials and methods in some cases. There is so many of these errors in this manuscript. For example: in page 3 line 89, "This group of 2,728 DE proteins sequences were submitted to analysisinthe90 BlastKOALA annotation tool for K number assignment". In the results section, authors only need to write the results, or: "A global proteomics analysis was performed by a label-free quantitation approachusing118 spectral counting generated by LC-MS/MS. Proteins were identified and quantifiedusing119 PatternLab V following the criteria mentioned in the Materials & Methods section". These sentences must be deleted or included in material and methods.

Response: Thank you for your comment. We accepted your suggestion and made several changes in the Results section (Please, see the revised version of the MS).

Therefore, I regret to say this manuscript in this form is not suitable for publication in Plants unless there are fundamental changes in it.

Response: Please, see if the revised version of the MS as it is became suitable for publication in Plants.

.

Reviewer 3 Report

The paper is very relevant and important in regard of saving tropical rainforest.  Methods are adequate, conclusions are well based on the results.

However:

Lines 34, 41, 46 etc, correct the size of “y” letter.

Lines 357,358. Define here again the plant material used, regardless if it was already in the cited literature.

Author Response

Response to Reviewers

Reviewer 01.

The MS entiled "Insights from a Multi-Omics Integration (MOI) study in oil palm (Elaeis guineensis Jacq.) response to abiotic stresses: Part one – Salinity is well written and may be accepted for publications after the below revisions.

P.1-L.14-15 please re-phrase the sentence.

Response: Thank you for your comment. Done! Please, see revised version of the MS.

Introduction:

What are the translational value of omics, please add at least  para on this and correlate on salinity. Why omics important for salinity?

Response: Thank you for your comment. The use of the different omics platforms available nowadays, alone or via an integrative strategy, is allowing researchers everywhere to gain many more insights in the response of a plant to an environmental stress than before the "omics era". The MOI strategy - something that we can say is still in its early stage - will definitely facilitate the hypothesis generation leading to an even deeper understanding of the functions and mechanisms behind the plant x environment interactions. Salinity is just one of the many environmental stresses, which understanding is gaining so much from this new era. We see what you want to achieve when you ask us to add a paragraph about the translational value of omics, correlating with salinity. However, we decided not to do so here in this MS. As you can see from the title of this MS, this is only the first chapter we wrote applying MOI to study the response of oil palm to abiotic stresses. As second chapter (Part two - Drought) is already been written, and there we believe is the best moment to stress the translational value of omics. Based on that, I kindly ask you to accept our response as it is now to your first comment. All other comments and questions you presented were accepted and resulted in important modifications in the MS (Please, see the revised version of the MS).

Please provide the pathway (KEGG).

Response: Thank you for your comment. We (the authors) accepted your suggestion and added the KEGG numbers for each one of the pathways mentioned (Please, see the revised version of the MS).

How much do metabolites or genes downregulate and upregulate? Please provide the table in the MS. Kindly add in Abstract section and also in results.

Response: Thank you for your question. We accepted your suggestion and added two new columns to Table 3, where you are going to find the False Discovery Rate (FDR) and the Log2(FC) from the differentially expressed metabolites. Regarding the genes/proteins differentially expressed in the transcriptomics part of the study, they are provided in the Supplemental Table S1. Besides, we also added this information in the Abstract and Results sections (Please, see the revised version of the MS).

I could not see the figs in the MS.

Response: Thank you for your comment. We the authors thought that the reviewers would have access to the figures, tables, and supplementary material submitted together with the main document; but apparently that was not the case. So, we did add all figures and tables to the revised version of the MS. We hope that this will help you in your evaluation of this MS.

 P.4  L. 128-129: 62 and 89 proteins were uniquely detected in control and stressed samples, respectively (Figure 3a, Supplementary Table S3).

Kindly  provide the important "20" stressed proteins in the table on the basis of fold changes and p value. The remaining can be included in the supplementary file.

Response: Thank you for your comment. A new table was added to the MS (Table 3 in the revised version of the MS). This table shows the 20 differentially expressed proteins detected in both biological conditions (Stressed and Control) with statistical significance (FDR ≤ 0.05). Please, see the revised version of the MS.

In methods:

In 4.1 W-47.925°)..    delete  ………………(.)

Response: Thank you for your comment. Done! Please, see revised version of the MS.

Reviewer 02.

In the material and method section authors have used 0.0 to 2 g of NaCl per 100 g of substrate)

The questions are that:

1- What do they mean by substrate? It must be explained more clearly and in details.

Response: Thank you for your question. We (the authors) modified the text to explain more clearly and in details, as requested. The new text in the revised version of the MS is: "The experiment consisted of five salinity levels (0.0, 0.5, 1.0, 1.5, and 2.0 g of NaCl per 100 g of substrate [a mixture of vermiculite, soil, and the Bioplant commercial substrate (Bioplant Agrícola Ltda., Nova Ponte, MG, Brazil), in a 1:1:1 ratio, on a dry basis], with four replicates in a completely randomized design (for additional details, see: [6]). The substrate mixture was fertilized using 2.5 g L-1 of the N-P2O5-K2O formula (20-20-20). For the omics (transcriptomics, metabolomics, and proteomics) analysis described in the present study, we collected the apical leaves from control and stressed plants (0.0 and 2.0 g of NaCl per 100 g of substrate) 12 days after imposition of the treatments (DAT)."

2- On what basis do the authors claim that these concentrations are very high salinity stress for this plant?

Response: Thank you for your question. The study described in this MS used transcriptomics, metabolomics, and proteomics data collected from leaves obtained in a previous study (Vieira et al., 2020; or [6]). In that previous study, we did measure the electrical conductivity (EC) for all treatments tested (see Figure 2 in Vieira et al., 2020). In the stress treatment (2.0 g of NaCl per 100 g of substrate), the EC is about 40 dS m-1. This level of salinity is then 10 times higher than the minimal level in a soil for it to be considered saline (EC > 4 dS m-1). So, this level of salinity is very high not only to oil palm (Elaeis guineensis), but also to any terrestrial plant species.

To help on linking these two studies, Vieira et al. (2020) and the MS, we modified the last paragraph in the introduction section. Please, see the revised version of the MS.

Tables and figures are missing in the manuscript and without them it is not possible to make a correct judgment. Authors should include relevant tables and figures in a comprehensible way for readers in their manuscript. Therefore, the whole manuscript are descriptive results. It is understandable that how SOA, and MOI analysis of the metabolome, transcriptome, and proteome profiles show differences among salinity levels.

Response: Thank you for your comment. We thought that the reviewers would have access to the figures, tables, and supplementary material submitted together with the main document; but apparently that was not the case. So, we did add all figures and tables to the revised version of the MS. We hope that this will help you in your evaluation of this MS.

It is not correct to cite the references in the result section. The authors must cite the reference in the discussion section. For example, see page 3, line 85:

The results are written in the style of materials and methods in some cases. There is so many of these errors in this manuscript. For example: in page 3 line 89, "This group of 2,728 DE proteins sequences were submitted to analysisinthe90 BlastKOALA annotation tool for K number assignment". In the results section, authors only need to write the results, or: "A global proteomics analysis was performed by a label-free quantitation approachusing118 spectral counting generated by LC-MS/MS. Proteins were identified and quantifiedusing119 PatternLab V following the criteria mentioned in the Materials & Methods section". These sentences must be deleted or included in material and methods.

Response: Thank you for your comment. We accepted your suggestion and made several changes in the Results section (Please, see the revised version of the MS).

Therefore, I regret to say this manuscript in this form is not suitable for publication in Plants unless there are fundamental changes in it.

Response: Please, see if the revised version of the MS as it is became suitable for publication in Plants.

Reviewer 03.

The paper is very relevant and important in regard of saving tropical rainforest.  Methods are adequate, conclusions are well based on the results.

However:

Lines 34, 41, 46 etc, correct the size of “y” letter.

Response: Thank you for your comment. We (the authors) are sorry, but we did not find the "y" letters of different sizes.

Lines 357,358. Define here again the plant material used, regardless if it was already in the cited literature.

Response: Thank you for your comment. We accepted your suggestion and made some changes in the 4.1. item of the Materials and Methods section (Please, see the revised version of the MS).

Round 2

Reviewer 1 Report

The MS has been revised accordingly.

Author Response

Dear Editors and Reviewers,

As requested, I revised the duplicated part in 4.3 Proteomics Data Analysis, according to the duplicate report sent to me by email. I highlighted in red the new text for this part.
To help with the final evaluation by the Reviewers and Editors, I did highlight all the modifications done in the text during the first and second revision exercises.
Thank you very much for your comments/suggestions, which helped us improve this MS.

Best regards,

Manoel Souza

Reviewer 2 Report

The authors have made significant improvements to their manuscript and have addressed most of the reviewer's concerns, so their manuscript in current status is recommended for publication in the Plants.

Author Response

(The authors gave the same response as above.)
